# Effect of autologous amniotic membrane and fluid on wound healing and complications of cesarean section: Study protocol of a factorial randomized controlled trial

Kasra Jafari[ID][1,2☯], Marzieh Vahid-Dastjerdi[3☯], SeyedAhmad SeyedAlinaghi[1,4], Amene Abiri[ID][3]*

**1** Research Development Center, Arash Women's Hospital, Tehran University of Medical Sciences, Tehran, Iran, **2** Department of Epidemiology, School of Public Health, Iran University of Medical Sciences, Tehran, Iran, **3** Department of Obstetrics and Gynecology, School of Medicine, Arash Women's Hospital, Tehran University of Medical Sciences, Tehran, Iran, **4** Iranian Research Center for HIV/AIDS, Iranian Institute for Reduction of High-Risk Behaviors, Tehran University of Medical Sciences, Tehran, Iran

☯ These authors contributed equally to this work.
* abiri@tums.ac.ir

## Abstract

### Background

Cesarean section rates have risen globally. While a cesarean section offers benefits, it is associated with complications like infection, fluid collection, and wound dehiscence. The psychological impact of cesarean section scars can also affect women's quality of life. Amniotic membrane and fluid have shown promise in enhancing wound healing due to their rich content of growth factors and nonimmunogenic properties. This study aims to assess the effects of autologous amniotic membrane and amniotic fluid on wound healing, surgical site infection, pain, and complications following cesarean section.

### Materials and methods

This controlled, randomized, double-blind (participants and the physician assessing outcomes are blinded), 2 × 2 factorial trial with a sample size of 228 is being conducted at Arash Women's Hospital, Tehran, Iran. It involves four groups: amniotic membrane dressing, amniotic fluid spray, a combination of both, and one group receiving standard of care, assessing the effects of autologous amniotic membrane and amniotic fluid on cesarean section wound healing, infection, pain, and complications. Primary outcome (wound healing) would be measured by the Patient and Observer Scar Assessment Scale. Secondary outcomes are surgical site infection, pain via the Visual Analog Scale, surgical wound dehiscence, seroma, hematoma, and bleeding. Results will be analyzed using two-way ANOVA and logistic regression.

**Data availability statement:** No datasets were generated or analysed during the current study. All relevant data from this study will be made available upon study completion.

**Funding:** The author(s) received no specific funding for this work.

**Competing interests:** No authors have competing interests.

**Abbreviations:** NVD, Natural Vaginal Delivery; CS, Cesarean Section; AM, Amniotic Membrane; AF, Amniotic Fluid; MD, Mean Difference; Hb, Hemoglobin; BMI, Body Mass Index; SD, Standard Deviation; VAS, Visual Analog Scale; POSAS, Patient and Observer Scar Assessment Scale; PSAS, Patient Scar Assessment Scale; OSAS, Observer Scar Assessment Scale; SSI, Surgical Site Infection; ANOVA, Analysis of Variance; FBS, Fasting Blood Sugar; OGTT, Oral Glucose Tolerance Test; VT, Vital Signs; CBC, Complete Blood Count.

## Discussion

This research protocol is the first to examine autologous amniotic membrane and amniotic fluid effects on cesarean section scar healing and complications. If effective, amniotic membrane and amniotic fluid could enhance CS scar aesthetics and reduce complications without extra cost.

## Trial registration

This trial was registered at the Iranian Registry of Clinical Trials (Registration number: **IRCT20220408054454N1**, Registration date: **2024-07-15**).

## Introduction

In the last three decades, CS increased from 6.7% to 19.1% globally [1]. It is especially a concerning issue in Iran, with a prevalence of 48% [2], leading some experts to call it a "cesarean epidemic" [3] considering it being nearly five times of 10% recommended by the World Health Organization (WHO) [4]. Considering its high prevalence, it seems necessary to study complications related to CS in Iranian women.

Although CS has many advantages, including protecting the perineum, eliminating the fear of labor pain (tokophobia), fulfilling the need to have authority, decreasing urinary incontinence, and painful perineum [5,6], still, the dangerous side effects of this operation threaten the life and health of mothers who undergo CS surgery. Like other surgeries, CS can be associated with surgical site complications, including infection (5.63% globally), fluid collection (hematoma and seroma, 2–5%), and dehiscence (2–7%) [7,8]. Moreover, the CS wound scar may seem like an inevitable aspect of the CS which is hidden beneath underwear and thus is not important, but it can heavily impact psychological health, with a qualitative study reporting "Don't like looking at scar" as the most prevalent effect of CS scar on its participants (31.5%) and other effects on clothing selection (27.1%), self-confidence (25.6%), altered relationship with partner (9.9%), and concerns about wound healing (9.4%) [9].

From the early 20th century, clinical applications of the human amniotic membrane (AM) for wound healing were introduced [10]. The studies show that AM is effective in enhancing healing (Standardized mean differences = −0.99 [Confidence interval 95% (CI 95%): −1.40, −0.58]) and reducing pain (Mean difference [MD] = −2.35 [CI 95%: −2.72, −1.98]), scarring (MD = −0.72 [CI 95%: −0.94, −0.50]), infection (Risk ratio = 0.51 [CI 95%: 0.28, 0.92]), and other subsequent wound complications [11–13]. The AM plays an important role in fetal development with its numerous growth factors, cytokines, and signaling molecules, which are also crucial in wound healing and tissue regeneration [14]. Furthermore, its nonimmunogenic nature makes it a good choice to be used as a matrix for cell deposition [15]. The amniotic fluid (AF) also has relatively similar biological characteristics and is known to enhance wound healing [16].

Autologous AM and AF outperform processed allogeneic products due to their lower cost, reduced immunogenicity, and enhanced safety profile. Unlike processed

alternatives, autologous AM (harvested during CS) requires no donor screening, sterilization, or long-term storage, significantly reducing expenses. Commercial products have higher costs because of production, regulatory compliance, and logistical handling, whereas autologous AM is immediately available at no additional cost for eligible patients. Immunologically, autologous AM eliminates rejection risks since it originates from the patient's own tissue, while processed allogeneic products may retain residual antigens despite sterilization, potentially triggering immune responses. Additionally, autologous AM avoids disease transmission concerns associated with unprocessed donor tissue, as it bypasses donor-to-recipient exposure. Processing methods can also diminish AM's bioactive properties (e.g., growth factors), further favoring autologous use when feasible. Based on the factors outlined above, we conclude that in the context of cesarean deliveries, autologous AM may represent a cost-effective, immunologically compatible, and clinically practical alternative to processed grafts.

While AM and AF both promote wound healing, they differ in key ways: AM provides a biologically active barrier rich in anti-inflammatory cytokines, growth factors, and extracellular matrix proteins that accelerate re-epithelialization, reduce scarring, and inhibit fibrosis [11]. AF, by contrast, contains pluripotent stem cells and signaling molecules in liquid form that may primarily apply paracrine effects on cell migration and proliferation [16]. To the best of our knowledge, no prior study has applied AM and AF separately or together to surgical incisional wounds; existing literature focuses overwhelmingly on chronic wounds such as diabetic ulcers, pressure ulcers, and burns [11,12,16,17]. Considering its benefits and high safety due to its autologous nature, we decided to conduct this study to investigate the possible effects of the AM and AF separately and in combination with each other in wound healing, surgical site infection (SSI), and pain of the patients undergoing CS.

## Materials and methods

### Study design and setting

This study is a controlled, randomized, double-blind, 2 × 2 factorial trial, implemented in the Arash Women's Hospital, Tehran University of Medical Sciences (TUMS), Tehran, Iran. The study evaluates two interventions (AM and AF), each applied alone or in combination, compared to a control group receiving neither. Patients meeting the eligibility criteria are entering the study and are followed until 28 days post-operative in different time steps. While the initial registration in the IRCT indicated a projected start date of 22/07/2024, the actual recruitment of participants began on 01/10/2024 due to administrative finalizations. We estimate that the participant recruitment will be completed by 31/10/2025. Data collection will be completed one month later on 30/11/2025, and results are expected to be ready by 31/12/2025. Fig 1 provides an overview of the study steps.

### Ethical approval

The protocol, written informed consent forms, recruitment materials, and all participant materials were approved by the Research Ethics Committee of the Tehran University of Medical Sciences (TUMS) and are in accordance with the ethical principles and the national norms and standards for conducting medical research in Iran (Approval ID: IR.TUMS. MEDICINE.REC.1402.758). Also, this trial was registered at the Iranian Registry of Clinical Trials (Registration number: **IRCT20220408054454N1**, Registration date: **2024-07-15**). Ethical approval, SPIRIT checklist, Farsi, and English institutional review board-approved research proposals are provided as S1-S4 Files.

### Outcome measures

**Primary outcome: Wound healing and the cosmetic results.** The primary outcome measure is the wound healing measured by the patient and observer scar assessment scale (POSAS) summary score, which consists of the Patient Scar Assessment Scale (PSAS) score and Observer Scar Assessment Scale (OSAS) score. The POSAS is a reliable and

---

*Prior to enrollment:*
Total N:104 . Obtain written informed consent. Screen potential participants by inclusion and
exclusion criteria; obtain history, document.

Randomization

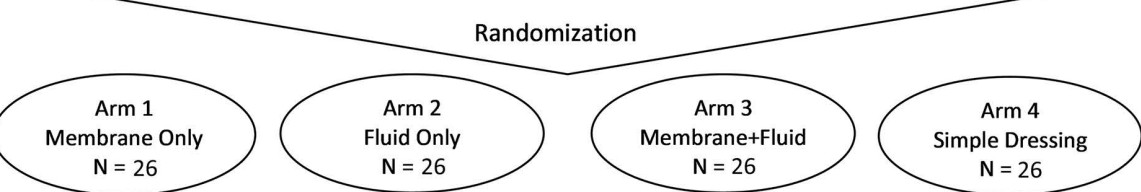

| Arm 1 Membrane Only N = 26 | Arm 2 Fluid Only N = 26 | Arm 3 Membrane+Fluid N = 26 | Arm 4 Simple Dressing N = 26 |

---

*Day -1: Before surgery*
Perform baseline assessments.
Demographic, socio-economic, obstetric and gynecologic characteristics, fasting blood sugar, oral
glucose tolerance test (OGTT), 2 hours OGTT, vital sings, complete blood count test, Sonography.

---

*Day 0: Surgery*
*Administertation of study intervention.*

---

*Day 0: 4, 12, and 24 hours post-operative*
Assess itchiness and irritation of the wound.
Pain severity score.
In the case of being administered, name, dosage, and route of administration of analgesic medicine.
Any adverse outcome.

---

*Day 0: 36 hours post-operative*
Assess itchiness, irritation, surgical site infection symptoms (localized pain or tenderness, localized
swelling, erythema, or heat), dehiscence, seroma, and hematoma of the wound.
Pain severity score.
In the case of being administered, name, dosage, and route of administration of analgesic medicine.
Any adverse outcome.

---

*Days 10 and 28 post-operative*
Assess itchiness, irritation, surgical site infection symptoms (localized pain or tenderness, localized
swelling, erythema, or heat), dehiscence, seroma, and hematoma of the wound.
Complete patient observer scar assessment scale (POSAS) questionnaire by phycian and patient.
Suture removal day.
Pain severity score.
In the case of being administered, name, dosage, and route of administration of analgesic medicine.
Any adverse outcome.

---

*Final Assessment*
Use two-way ANOVA and logistic regression for analysis.
After sample size reaches 50% perform interim analysis to detect differences with a p-value < 0.001

---

**Fig 1. Overview of the study.**

valid scar assessment scale designed for a subjective evaluation of various types of scar formation from the perspectives of both the participant and the clinician [18]. The authors will be using POSAS 3.0 for this study [19].

The Patient Scale includes 17 items, each rated on a 5-point Likert scale ranging from "not at all" to "extremely," evaluating various aspects of the scar over the past week and at the present moment. These items cover parameters such as scar color, pain, sensitivity, itching, surface texture, elevation, dryness, widening, and other sensory symptoms like burning or tingling. The scale is intended to reflect the patient's subjective experience of the scar, allowing for a more holistic evaluation of scar quality beyond clinical observation alone. The Observer Scale, completed by a trained clinician, assesses additional morphological and functional characteristics of the scar and contains 9 items. The minimum scores for PSAS and OSAS are 17 and 9, and the maximum scores are 85 and 45, respectively. No validated cut-off values exist for POSAS 3.0. It is explicitly intended as a continuous outcome measure for scar quality (higher = worse scar), not a diagnostic test.

The Persian version of the POSAS was translated, culturally adapted, and validated by Naghizadeh et al. in a study of 60 post-surgical patients. It demonstrated strong content validity, high internal consistency (Cronbach's alpha: 0.88 for the Observer Scale and 0.81 for the Patient Scale), and good test-retest reliability, confirming its suitability for clinical and research use in Iran [20].

Participants and a general practitioner (or an obstetrician and gynecologist) complete the scales at 10 and 28 days after surgery.

**Surgical Site Infection (SSI).** We defined the SSI by the Centers for Disease Control and Prevention (CDC) guideline for surgical site infection events [21]. The participants will be visited by a general practitioner (or an obstetrician and gynecologist) for purulent drainage from the superficial incision, localized pain or tenderness, localized swelling, erythema, or heat. The specimen is obtained aseptically from confirmed cases of the SSI and is sent to the laboratory for microbiological testing. Patients with SSI are categorized as superficial incisional SSI, deep incisional SSI, and organ/space SSI based on the mentioned guidelines. Incidence of the SSI is assessed on 36 hours, 10, and 28 days postoperative.

**Pain.** Surgical wound pain is assessed using a visual analog scale (VAS) ranging from one to ten. A general practitioner (or an obstetrician and gynecologist) asks the participant about her pain at the four, 12, 24, 36 hours, 10, 28 days postoperative. Also, analgesic medication use by name of the medication, route of intake (oral, intramuscular, intravenous), dosage, and frequency are recorded at the aforementioned time steps.

**Other surgical site complications.** All the participants are assessed by a general practitioner (or an obstetrician and gynecologist) at 36 hours, 10, and 28 days postoperative for surgical wound dehiscence, seroma, hematoma, and bleeding.

**Safety and adverse event reporting.** All participants are monitored intraoperatively and postoperatively for any signs of adverse events, including but not limited to allergic and immunologic reactions. Any unexpected or serious adverse events are documented and reported in accordance with institutional protocols. In cases of suspected allergic or immunologic reactions, immediate clinical evaluation and appropriate management will be initiated. All adverse events are recorded in the case report forms (CRFs) and reviewed by an independent safety officer not involved in the study procedures.

## Procedure

In brief, all the assessments and evaluations are presented in the Table 1.

**Recruitment and baseline assessment.** All the participants are recruited from clients referred to Arash Women's Hospital. The researchers recruit participants through in-person explanation, and interested clients receive detailed information about the study. After checking that the eligibility criteria have been met, a written informed consent will be signed by the participant. The baseline assessment involves demographic characteristics (age, weight, height, and

# Table 1. Project assessment and time steps.

| TIMEPOINT | Enrolment Day −1 | Allocation 0 | Intervention Day 0, 0 hour | Study Visit 1 Day 0, 4 hours postoperative | Study Visit 2 Day 0, 12 hours postoperative | Study Visit 3 Day 0, 24 hours postoperative | Study Visit 1 Day 1, 36 hours postoperative | Study Visit 5 Day 10 ±2 days | Final Study Visit 6 Day 28 |
|---|---|---|---|---|---|---|---|---|---|
| **STUDY PERIOD** | | | **Post-allocation** | | | | | | **Close-out** |
| **ENROLMENT:** | | | | | | | | | |
| Eligibility screen | x | | | | | | | | |
| Informed consent | x | | | | | | | | |
| Allocation | | x | | | | | | | |
| **INTERVENTIONS:** | | | | | | | | | |
| Autologous AM | | | x | | | | | | |
| Autologous AF spray | | | x | | | | | | |
| Autologous AM and AF spray | | | x | | | | | | |
| Standard of care | | | x | | | | | | |
| **ASSESSMENTS:** | | | | | | | | | |
| Medical history and physical examination | x | x | | | | | | | |
| Demographics | x | | | | | | | | |
| Vital signs | x | | | | | | | | |
| Hematology and serum chemistry tests | x | | | | | | | | |
| POSAS questionnaire | | | | | | | | x | x |
| Surgical site infection assessment | | | | ←——————————————————————→ | | | | | |
| Wound complication assessment | | | | ←——————————————————————→ | | | | | |
| Wound pain assessment | | | | ←——————————————————————→ | | | | | |
| Analgesic medication intake | | | | ←——————————————————————→ | | | | | |
| Adverse event review and evaluation | | | | ←——————————————————————→ | | | | | |

AM: amniotic membrane, AF: amniotic fluid, POSAS: patient and observer scar assessment scale.

nationality), socioeconomic status (education and place of residence), and medical history and conditions (parity, number of former CS, fasting blood sugar (FBS), two hours oral glucose tolerance test (OGTT), complete blood count (CBC) test, vital signs (VT), and history of underlying diseases and comorbidities) of the participants. All baseline assessments will be conducted by trained medical residents.

**Eligibility criteria.** All women aged 18–45 years, singleton pregnancy, having a previous plan for CS and elective CS, planned gestational age of at least completed 36 weeks (≥37 + 0 weeks) at the time of delivery, with hemoglobin (Hb) above 10 g/dL at the time of hospitalization, able to provide the consent, and with intention to be available for the entire research period and to complete all relevant study steps, including follow-up visits and phone calls are included in the study.

The researchers exclude all participants with a body mass index (BMI) greater than or equal to 40 at the time of entering the study, placenta previa or placenta accreta, history of intestinal or urological surgery, multiple pregnancy, known or suspected immunodeficiency (including HIV, hepatitis B or C infection), use of immunosuppressive drugs, chemotherapy, radiotherapy, or taking systemic corticosteroids in the 4 weeks before surgery, known tobacco or drug use, history of keloid formation, participation in another clinical trial in the last three months, any serious cardiovascular, pulmonary, hepatic, renal, digestive, hematologic, endocrine, metabolic, neurological and psychological diseases, frequent vaginal examinations, emergency CS, chorioamnionitis or other systemic infections at the time of presentation for CS, including evidence of subabdominal skin infection (e.g., fungal infection), and any condition that in the opinion of the investigator may pose a risk to the health of the participant or interfere with the evaluation of the study participants.

**Intervention.** The participants are divided into four groups:

- Intervention group A: dressing with autologous AM for CS wound.

- Intervention group B: dressing with autologous AF spray for CS wound.

- Intervention group C: dressing with autologous AM and AF spray for CS wound.

- Control group D: simple dressing for CS wound (standard of care).

For all participants in intervention group A, after the surgery, a portion of the AM corresponding to the size of the surgical incision is removed, washed with sterile normal saline, and placed on the incision site before applying the usual dressing. The autologous AM is stored in a sterile receiver filled with sterile normal saline at all times between harvest and application. The storage duration varies depending on the length of the surgery, typically not exceeding 30–60 minutes, and the storage temperature is maintained at room temperature (approximately 20–25°C). In intervention group B, the extracted autologous AF is sprayed directly into the skin incision immediately before wound closure. In group C, both interventions are applied. In the control group D, subjects receive standard care with a simple dressing. Wound closure was standardized across all groups and performed using absorbable subcutaneous sutures followed by non-absorbable interrupted sutures for the skin, applied by the same surgical team following a consistent technique. During the surgery, after the uterine incision, the following steps are performed for groups B and C:

1. After incising the lower part of the uterus, the amniotic sac is opened with an amniotic perforator.

2. A 10 ml syringe is used to drain the AF.

3. Then, the 10 ml syringe containing the AF is observed for contamination (for example, the presence of meconium), and then the AF is sprayed throughout the skin incision (1.5 to 2.0 ml).

4. Then, the cut is dressed as usual.

**Randomization.** To make sure that the same number of participants is allocated to all four groups, block randomization with block sizes of 4, 8, and 12 is used. A randomized list is generated using the Sealed Envelope website

[22]. The number of blocks and the allocation sequence in the randomized list are only available to an external observer responsible for participant allocation, and the researchers are not aware of it. This external observer will not have a role in analyzing the data, assessing, or evaluating the participants. Adherence to the randomization protocol will be closely monitored throughout the trial.

**Blinding and study termination.** The wound dressings of all four groups are similar from the outside, and since the AM is placed under the dressing, the subject will not be aware of her allocation. Blinding of the surgical team is not possible. The physician who examines the condition of the wound is not aware of the participant's group, and the participant's wound dressing is opened by another physician or nurse before the examination by the physician, so that the examining physician does not know about the assignment. In general, except for the surgical team and the external observer, all participants involved in the assessment and analysis of the study will be blinded.

The process of unblinding and/or discontinuing a participant will be done only in case of necessity and the occurrence of a serious side effect (with the diagnosis and order of the supervisor of the study). The blinding of the study will be broken only in cases where it is not possible to manage and treat the participant appropriately and adequately without knowing the type of intervention she received.

**Concealment.** Sequentially numbered, opaque, sealed envelopes are used for concealment. Each envelope is identical, fully opaque to prevent visibility of contents, and sealed to prevent tampering. Envelopes are stored in a locked cabinet and are only accessible to an independent research assistant not involved in participant recruitment or assessment. Envelopes are opened sequentially only after confirming participant eligibility and obtaining informed consent.

## Sample size

According to the study of Cromi et al. [23] who investigated the healing and pain of CS wounds in four different suture methods, the sample size was calculated. The common method for CS wound sutures in Arash Women's Hospital is to use absorbable monofilament sutures in the subcutaneous layer of the skin and to use nylon monofilament sutures in the skin itself. According to the study, the score of the patient part for the POSAS questionnaire was 19.3 (standard deviation [SD] = 7.5), and for the observer part, 20.6 (SD = 7.6). We powered the study to detect a 25% reduction in the POSAS score:

$$\textit{Cohen's d (Based on the patient score)} = \frac{\Delta}{\sigma} = \frac{0.25 \times 19.3}{7.5} \cong 0.6433$$

$$\textit{Cohen's d (Based on the observer score)} = \frac{\Delta}{\sigma} = \frac{0.25 \times 20.6}{7.6} \cong 0.6776$$

For a factorial trial, the main effect (e.g., AM present vs AM absent) is estimated by pooling the two cells with AM present vs the two cells without AM. A two-group comparison with effect $d$ corresponds to an ANOVA effect size $f = d/2$:

$$f\ (\textit{Based on the patient score}) = \frac{d}{2} = \frac{0.6433}{2} = 0.3217$$

$$f\ (\textit{Based on the observer score}) = \frac{d}{2} = \frac{0.6776}{2} = 0.3388$$

We calculated the sample size using G*Power 3.1.9.7 considering an F test for main effects (ANOVA: fixed effects, main effects and interactions) with $f$ (Based on the patient score) = 0.3217 and $f$ (Based on the observer score) = 0.3388,

α = 0.05, 1 − β = 0.90 (two-sided), and numerator degrees of freedom = 1, the highest required total sample size is 104 (≈26 participants per cell; i.e., four cells).

Since the highest sample size obtained was 26, taking into account a 10% dropout, the final number for each cell was calculated at 29 participants (in total, 116 participants).

## Statistical analysis

Analyses will be conducted using Stata 17.0 (StataCorp, College Station, TX, USA) on an intention-to-treat basis. This is a 2 × 2 factorial trial with two interventions (AM and AF; present vs absent) producing four groups. Main effects will be estimated by pooling across the relevant cells (AM present = AM only + AM + AF vs AM absent = AF only + control; AF present = AF only + AM + AF vs AF absent = AM only + control), and the AM × AF interaction will also be tested. Normality of continuous outcomes will be assessed using the Shapiro–Wilk test, histograms, and Q-Q plot.

The primary outcome is the POSAS score at day 28, which represents the final cosmetic and functional quality of wound healing. The POSAS score at day 10 will also be collected, because patients routinely return for suture removal at that time, allowing early evaluation and minimizing potential loss to follow-up if some participants do not attend the day-28 visit. Since there are only two time points and the clinical focus is on the final outcome, a longitudinal analysis is not expected to provide additional value or sufficient power. Therefore, the primary analysis will compare day-28 POSAS scores among groups using two-way ANOVA (factors: AM, AF, and AM × AF). If model assumptions are violated, a generalized linear model or a rank-based factorial ANOVA will be applied.

The primary safety outcome is the incidence of SSI within 28 days after cesarean section. SSI will be assessed at 36 hours, day 10, and day 28 only to detect any occurrence of infection; the timing of onset is not clinically central. Therefore, the cumulative incidence of SSI by day 28 will be analyzed using logistic regression, including AM, AF, and their interaction.

Postoperative pain, measured using VAS score at 4, 12, 24, and 36 hours, day 10, and day 28, will be analyzed as a repeated-measures outcome. We will perform a linear mixed-effects model (LMM) including fixed effects for time, AM, AF, and their interactions (AM × AF, time×group). However, if the sample size is insufficient for stable model estimation, we will instead report per-timepoint results using two-way ANOVA (factors: AM, AF, and AM × AF). This fallback approach allows assessment of group differences at each time while acknowledging that the study is underpowered for small time×group interactions.

The trial is powered for the main effects of AM and AF at the prespecified final endpoints (POSAS at day 28 and SSI by day 28). If the interaction term is not significant (p ≥ 0.05), primary emphasis will be placed on the main effects. Significant interactions will be explored with Bonferroni-adjusted pairwise comparisons. Interim analysis will occur after 50% recruitment with p < 0.001; final analyses will use p < 0.05 (two-sided).

## Discussion

To the best of our knowledge, this is the first study that investigates the healing effects of autologous AM and AF separately and combined on the CS scar. The authors also aim to assess the effects of the AM and AF on CS incision complications, such as pain, SSI, and other prevalent complications like surgical wound dehiscence, seroma, hematoma, and bleeding.

This protocol has three key strengths. Firstly, it is a factorial randomized clinical trial with a control group receiving standard of care, and its design does help us to find out the cumulative effects of AM and AF use besides their single use. Secondly, the autologous nature of the AM and AF used makes the investigated interventions both safe and low-cost. Finally, the primary outcome of the study (scar wound healing) is assessed from the perspectives of both the participant and clinician. A limitation of this study is its single-center design, which may limit the generalizability of the findings to other

settings. Additionally, the follow-up period was relatively short, restricting the assessment of long-term outcomes such as scar maturation and delayed complications. Due to the nature of the interventions, blinding of the surgical team is not feasible, which may cause performance bias.

The use of autologous AM and AF in this study demonstrates potential feasibility even in low-resource settings. Since both AM and AF are derived from the patient's own body at the time of CS, there is no need for external donor tissue, costly processing, or long-term storage facilities. The preparation involves minimal equipment—sterile containers, saline solution, and standard surgical instruments—and can be performed within the routine sterile environment of an operating room. This makes the approach relatively simple, low-cost, and adaptable for use in hospitals with limited resources, provided that proper sterile techniques and basic training are ensured.

In conclusion, if these interventions prove to be effective, patients undergoing CS will benefit from meaningful improvement in scar appearance and postoperative outcomes at no additional cost.

## Trial status

Recruitment has been initiated and is ongoing.

## Supporting information

**S1 File. Ethical approval.** IRB's ethical approval letter.
(PDF)

**S2 File. SPIRIT checklist.** Completed SPIRIT checklist.
(DOCX)

**S3 File. IRB-approved protocol.** Proposal file in Farsi, which was approved by the IRB.
(DOCX)

**S4 File. IRB-approved protocol English.** The proposal file, which was approved by the IRB translated into English.
(DOCX)

## Acknowledgments

The authors would like to thank the Arash Women's Hospital staff for their help.

During the preparation of this work, the authors used ChatGPT (developed by OpenAI, 2023) in order to improve the readability and language of this paper. After using this tool, the authors reviewed and edited the content as needed and take full responsibility for the content of the published article.

## Author contributions

**Conceptualization:** Kasra Jafari, Marzieh Vahid-Dastjerdi, SeyedAhmad SeyedAlinaghi, Amene Abiri.

**Investigation:** Kasra Jafari, Marzieh Vahid-Dastjerdi, SeyedAhmad SeyedAlinaghi, Amene Abiri.

**Methodology:** Kasra Jafari, Marzieh Vahid-Dastjerdi, SeyedAhmad SeyedAlinaghi, Amene Abiri.

**Project administration:** Marzieh Vahid-Dastjerdi, Amene Abiri.

**Supervision:** Amene Abiri.

**Visualization:** Kasra Jafari.

**Writing – original draft:** Kasra Jafari, Marzieh Vahid-Dastjerdi, SeyedAhmad SeyedAlinaghi, Amene Abiri.

**Writing – review & editing:** Kasra Jafari, Marzieh Vahid-Dastjerdi, SeyedAhmad SeyedAlinaghi, Amene Abiri.

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
