## [Decision Letter · Decision Letter 0]

15 Jul 2025

Dear Dr. Abiri,

Thank you for submitting your manuscript to PLOS ONE. After careful consideration, we feel that it has merit but does not fully meet PLOS ONE’s publication criteria as it currently stands. Therefore, we invite you to submit a revised version of the manuscript that addresses the points raised during the review process.

**ACADEMIC EDITOR: Please respond to all reviewers comments**

We look forward to receiving your revised manuscript.

Kind regards,

Ahmed Mohamed Maged, MD

Academic Editor

PLOS ONE

Journal Requirements:

5. In the online submission form, you indicated that the datasets used and/or analyzed during the current study will be available from the corresponding author upon reasonable request.

6. Please amend your list of authors on the manuscript to ensure that each author is linked to an affiliation. Authors’ affiliations should reflect the institution where the work was done (if authors moved subsequently, you can also list the new affiliation stating “current affiliation:….” as necessary).

7. Please include a copy of Table 1 which you refer to in your text on page 6.

8. Please remove all personal information, ensure that the data shared are in accordance with participant consent, and re-upload a fully anonymized data set.

Reviewers' comments:

Reviewer's Responses to Questions

**Comments to the Author**

1. Does the manuscript provide a valid rationale for the proposed study, with clearly identified and justified research questions?

Reviewer #1: Yes

Reviewer #2: Yes

Reviewer #3: Partly

2. Is the protocol technically sound and planned in a manner that will lead to a meaningful outcome and allow testing the stated hypotheses?

Reviewer #1: Partly

Reviewer #2: Yes

Reviewer #3: No

3. Is the methodology feasible and described in sufficient detail to allow the work to be replicable?

Reviewer #1: No

Reviewer #2: Yes

Reviewer #3: Yes

4. Have the authors described where all data underlying the findings will be made available when the study is complete?

Reviewer #1: Yes

Reviewer #2: Yes

Reviewer #3: No

5. Is the manuscript presented in an intelligible fashion and written in standard English?

Reviewer #1: Yes

Reviewer #2: Yes

Reviewer #3: Yes

You may also provide optional suggestions and comments to authors that they might find helpful in planning their study.

Reviewer #1: Thank you for providing this opportunity to review this manuscript. Please see my comments as follows:

Background

1. Authors should start “Background” directly from cesarean section and its consequences such as wound infection.

2. LINES 66-71 are redundant as the focus of this study is noy\t why women choose cesarean. I recommend Combine redundant CS epidemiology paragraphs (lines 53-75 and 27-33).

3. Add quantitative data on AM's efficacy from recent meta-analyses (e.g., Zelen et al. 2020).

4. Clarify why autologous AM/AF may outperform processed products (cost/immunogenicity).

5. In my opinion the following are not advantages of cesarean section and they are disadvantages of C/S: shorter recovery, shorter length of hospitalization, more successful breastfeeding, and control over the birth process.

6. What is the meaning of “delicense” in line 78? I think it should have been “dehiscence”.

Methods

1. As the researchers are not blind, this study is not double blind, and this is single blind.

2. authors should explain the POSAS scale in more detail. How many questions does it have? Who validated this scale in Iran?

3. Are the patients hospitalizing until 36 hours after C/S, if not, how authors evaluate the pain of participants in this time?

4. Add screening flowchart (CONSORT-style) showing enrollment numbers at each stage.

5. Specify who conducts baseline assessments (trained nurses? residents?).

6. Line 159: Why authors consider gestational age of 36 weeks, that is not term pregnancy.

7. Line 160: Hb: 10g/dL is considered anemia and it can deteriorate wound healing.

8. A randomized list is generated using the sealed envelope is not correct and needs revision.

9. Line 220: the explanation of allocation concealment is not enough.

10. please explain Storage duration/temperature for AM/AF and standardized wound closure technique across groups.

11. Add Safety: Add protocol for adverse event reporting (e.g., allergic reactions).

Discussion

1. Comparative Context: Contrast with similar AM studies in non-CS wounds.

2. Implementation: Address feasibility of AM/AF use in low-resource settings.

3. Limitations: Acknowledge single-center design and short follow-up.

Reviewer #2: Dear Editor,

PLOS ONE

Thank you also for the opportunity to contribute to the peer review process for manuscript titled: “Effect of Autologous Amniotic Membrane and Fluid on Wound Healing and Complications of Cesarean Section: Study Protocol of A Factorial Randomized Controlled Trial”.

This is a well-written manuscript on an important topic. I commend the researchers and wish them success. I commend the researchers and wish them success. It is a valuable contribution to the existing literature and will undoubtedly stimulate further discussion and investigation. The methodology is sound I encourage the authors to continue their work in this area and look forward to seeing the future fruits of their labors.

Best regards

Reviewer #3: The protocol here is for a 2x2 factorial trial of two related interventions - autologous amniotic fluid, and amniotic membrane.

The assumption in the factorial design is that the tow treatments are basically additive (there is no interaction between them as measured when one performs analysis of one intervention stratified by another). This could usefully be justified here in light of the statements on lines 90-1 where it might be thought that similar biological characteristics lead to sub-additive properties (e.g. taking 2 doses of paracetamol instead of 1 does not double the headache cure rate). Additionally, if the wound healing properties are well-known there is presumably good randomised evidence already - so the precise need for this trial needs to be made clear - where is the uncertainty which underpins ethical randomisation?

It would be better to describe this as a 2x2 design and not a four arm study of 3 interventions and 1 control - there are two interventions each of which is given or withheld.

The interventions do not sound blinded - how are the sprays applied and should one not have a dummy spray - the preparation of the interventions seems to be different. How is one to avoid performacne bias in the period before the assessment by a blinded observer.

I think the statistician needs to be aware of allocation (line 211) to analyse the data.

There is insufficient guarding of allocation concealment given the use of blocking in a single centre. Closed envelopes do not provide good concealment (see the excellent example in Altman's book on medical statistics) so how will this be policed?

If the pain on a VAS is 8 with sd 1.7 it is not normally distributed - how was this taken into account in the sample size calculation.

To allow for 10% dropout one must inflate by 1/(0.9) - so the sample size would need to be slightly more than 56 - as it is 56 does not allow for 10% dropout but rather only about 9%. Also this is two 2-way comparisons - so if one wants to power each intervention on the differences seen you need only 104 patients in total in each comparison. In other words the sample size is out by a factor of two since every patient in a factoria ldesign counts double.

The analyses are not describedf in suffiicent detail or in a fashion appropriate to a factorially designed trial.

**Do you want your identity to be public for this peer review?** For information about this choice, including consent withdrawal, please see our Privacy Policy

Reviewer #1: **Yes: ** Parvin Abedi

Reviewer #2: **Yes: ** Fereshteh Behmanesh

Reviewer #3: No

---

## [Author Response · Author response to Decision Letter 1]

12 Aug 2025

We sincerely appreciate the time and effort you and the reviewers have dedicated to evaluating our manuscript, "Effect of Autologous Amniotic Membrane and Fluid on Wound Healing and Complications of Cesarean Section: Study Protocol of A Factorial Randomized Controlled Trial" (Manuscript ID: PONE-D-25-30795). We are grateful for the constructive feedback, which has helped us improve the quality of our work.

We have carefully addressed all the comments and suggestions provided by the reviewers and uploaded it as a Word file (Response 2025-08-10.docx) in the submission portal.

---

## [Decision Letter · Decision Letter 1]

1 Sep 2025

Dear Dr. Abiri,

Thank you for submitting your manuscript to PLOS ONE. After careful consideration, we feel that it has merit but does not fully meet PLOS ONE’s publication criteria as it currently stands. Therefore, we invite you to submit a revised version of the manuscript that addresses the points raised during the review process.

We look forward to receiving your revised manuscript.

Kind regards,

Ahmed Mohamed Maged, MD

Academic Editor

PLOS ONE

Journal Requirements:

Additional Editor Comments:

**Please respond to all reviewer comments**

Reviewer's Responses to Questions

**Comments to the Author**

1. Does the manuscript provide a valid rationale for the proposed study, with clearly identified and justified research questions?

Reviewer #3: Partly

2. Is the protocol technically sound and planned in a manner that will lead to a meaningful outcome and allow testing the stated hypotheses?

Reviewer #3: Partly

3. Is the methodology feasible and described in sufficient detail to allow the work to be replicable?

Reviewer #3: No

4. Have the authors described where all data underlying the findings will be made available when the study is complete?

Reviewer #3: Yes

5. Is the manuscript presented in an intelligible fashion and written in standard English?

Reviewer #3: Yes

You may also provide optional suggestions and comments to authors that they might find helpful in planning their study.

Reviewer #3: Thank you for your response to my previous comments.

I note now that this is billed as a factorial design - so the power is different, and the calculation needs to show what the actual power is here for the design as chosen.

Please note also that you cannot analyse a factorial design given A vs B vs C vs D - it is okay not to know what is treatment and placebo (ie block blinding, not true blinding) but the two interventions need to be given separately to allow for the analyses proposed.

**Do you want your identity to be public for this peer review?** For information about this choice, including consent withdrawal, please see our Privacy Policy

Reviewer #3: No

---

## [Author Response · Author response to Decision Letter 2]

11 Sep 2025

Dear Dr. Ahmed Mohamed Maged,

We sincerely appreciate the time and effort you and the reviewer have dedicated to evaluating our manuscript, "Effect of Autologous Amniotic Membrane and Fluid on Wound Healing and Complications of Cesarean Section: Study Protocol of A Factorial Randomized Controlled Trial" (Manuscript ID: PONE-D-25-30795). We are grateful for the constructive feedback, which has helped us improve the quality of our work.

We have carefully addressed all the comments and suggestions provided by the reviewer. Below, we provide a point-by-point response to each remark, detailing the revisions made in the revised manuscript. All changes have been highlighted in the revised version for easy reference by yellow color.

Reviewer #3:

We sincerely thank you for your insightful and detailed comments. Your suggestions were very important in strengthening our methods section, and we greatly appreciate the time and expertise you dedicated to improving the quality of our manuscript.

In case if you believe that our amnedments do not fully satisfy your comments, please provide us with the exact sample size calculation formula (or software) and any specific statistical analysis technique you consider suitable for our methodology.

Comment 1: "I note now that this is billed as a factorial design - so the power is different, and the calculation needs to show what the actual power is here for the design as chosen.”

Response 1: Thank you for your detailed comment. We acknowledge the need to present the sample size in the context of a 2×2 factorial design. We have recalculated and clarified the sample size justification for the factorial design and added the full calculations to the revised manuscript (see revised Sample Size subsection). Briefly, the study has two factors (AM: present/absent; AF: present/absent). We powered the trial to detect a clinically meaningful 25% reduction in the POSAS patient score (same assumption used previously) (lines 272-290).

Comment 2: "Please note also that you cannot analyse a factorial design given A vs B vs C vs D - it is okay not to know what is treatment and placebo (ie block blinding, not true blinding) but the two interventions need to be given separately to allow for the analyses proposed.”

Response 2: Thank you so much. We agree that the statistical analysis must be factorial (main effects and interaction) rather than a simple A vs B vs C vs D comparison. We have revised the Methods to state explicitly that the main effects will be estimated by pooling across the relevant cells (AM present = groups A+C vs AM absent = groups B+D; AF present = B+C vs A+D), and that the AM × AF interaction will be tested by inclusion of the interaction term. We will report main effects and interaction statistics (with pairwise comparisons only if the interaction is significant) (lines 294-297 and 303-305).

Sincerely,

Dr. Amene Abiri

---

## [Decision Letter · Decision Letter 2]

30 Sep 2025

Dear Dr. Abiri,

Thank you for submitting your manuscript to PLOS ONE. After careful consideration, we feel that it has merit but does not fully meet PLOS ONE’s publication criteria as it currently stands. Therefore, we invite you to submit a revised version of the manuscript that addresses the points raised during the review process.

We look forward to receiving your revised manuscript.

Kind regards,

Ahmed Mohamed Maged, MD

Academic Editor

PLOS ONE

Journal Requirements:

Additional Editor Comments:

** Please respond to all reviewers comments** .

Reviewer's Responses to Questions

**Comments to the Author**

1. Does the manuscript provide a valid rationale for the proposed study, with clearly identified and justified research questions?

Reviewer #1: Yes

Reviewer #2: Yes

Reviewer #3: Yes

Reviewer #4: Partly

Reviewer #5: Yes

2. Is the protocol technically sound and planned in a manner that will lead to a meaningful outcome and allow testing the stated hypotheses?

Reviewer #1: Yes

Reviewer #2: Yes

Reviewer #3: Yes

Reviewer #4: Partly

Reviewer #5: Yes

3. Is the methodology feasible and described in sufficient detail to allow the work to be replicable?

Reviewer #1: Yes

Reviewer #2: Yes

Reviewer #3: Yes

Reviewer #4: Yes

Reviewer #5: Yes

4. Have the authors described where all data underlying the findings will be made available when the study is complete?

Reviewer #1: No

Reviewer #2: Yes

Reviewer #3: Yes

Reviewer #4: Yes

Reviewer #5: No

5. Is the manuscript presented in an intelligible fashion and written in standard English?

Reviewer #1: Yes

Reviewer #2: Yes

Reviewer #3: Yes

Reviewer #4: No

Reviewer #5: Yes

You may also provide optional suggestions and comments to authors that they might find helpful in planning their study.

Reviewer #1: Thank you for inviting me to review this manuscript. Please see my comments as follows:

Abstract

1. Please mention the primary outcomes and how did you measure all outcomes? What are the statistical tests?

Introduction

1. All of benefits that authors mentioned about cesaean section in comparison to normal vaginal delivery, in fact are the hazards of this surgery such as: shorter hospitalization, shorter recovery, better breastfeeding. Please revise.

2. Lines 102-103 needs to be cited.

3. Line 95: please write the reference.

Methods

1. What was the minimum and maximum score of POSAS? Is there any cut-off?

2. Statistical analyses: did authors attempt to report effect size?

Line 335: why authors wrote that human ethics and consent to participate is NA?

Reviewer #2: I commented before and accept this manuscript for publication. I believe it offers a significant contribution to the field, presenting compelling evidence and a well-reasoned argument. The methodology is sound, the data is meticulously analyzed, and the conclusions are logically derived. Furthermore, the writing is clear and concise, making the complex subject matter accessible to a wider audience. I commend the author(s) for their thorough research and insightful presentation.

Reviewer #3: Thank you for your updated protocol - please note however, that given the breakdown that is given in the response it is possible to unblind the statistician as for ghroups A,B,C,D you are telling us precisely which comparison they are in. So please remove the statement at line 252 as it is not correct.

Reviewer #4: Thanks to the authors of the manuscript

The manuscript needs corrections, so it is recommended that the following items be corrected:

1. The references in the text need to be updated.

4. The manuscript should be re-edited.

Reviewer #5: 1. Line 54: The phrase “… CS increased by 12.4% (from 6.7% to 19.1%)” should be edited. First, CS should be spelled out. Second, increased by 12.4% implies a relative increase (RRI), whereas the authors mean an absolute increase (ARI).

2. Lines 59–61: The reference used for the sentence is out of date (2012), and I think some of the mentioned advantages (for example, shorter recovery, shorter length of hospitalization, more successful breastfeeding, and control over the birth process) are not correct.

3. The expected recruitment start date registered in the IRCT (2024-07-22) differs from what is reported in the manuscript as the recruitment start date (01/10/2024). Please describe the reason in the protocol.

4. Line 35: It is mentioned that the trial is double-blind, but the person performing the intervention could not be blinded. According to CONSORT, please specify in the abstract which groups are blinded.

5. It is better not to use uncommon abbreviations such as “PO.”

6. Table 1: It is suggested to use “×” for each assessment time point instead of the current symbol.

7. Please report which number was used for the numerator degrees of freedom. I used 3, and the calculated sample size was 141.

8. Most of the outcomes will be measured more than once. Please explain how time and the interaction of time × group will be assessed during analysis.

9. Please describe where all data underlying the findings will be made available when the study is complete.

**Do you want your identity to be public for this peer review?** For information about this choice, including consent withdrawal, please see our Privacy Policy

Reviewer #1: No

Reviewer #2: No

Reviewer #3: No

Reviewer #4: No

Reviewer #5: No

---

## [Author Response · Author response to Decision Letter 3]

6 Oct 2025

Dear Dr. Ahmed Mohamed Maged,

We sincerely appreciate the time and effort you and the reviewer have dedicated to evaluating our manuscript, "Effect of Autologous Amniotic Membrane and Fluid on Wound Healing and Complications of Cesarean Section: Study Protocol of A Factorial Randomized Controlled Trial" (Manuscript ID: PONE-D-25-30795R2). We are grateful for the constructive feedback, which has helped us improve the quality of our work.

We have carefully addressed all the comments and suggestions provided by the reviewer. Below, we provide a point-by-point response to each remark, detailing the revisions made in the revised manuscript. All changes have been highlighted in the revised version for easy reference in yellow.

Reviewer #1:

We sincerely thank you for your insightful and detailed comments.

Comment 1: "Please mention the primary outcomes and how did you measure all outcomes? What are the statistical tests?”

Response 1: As suggested, in the “Abstract” we have now stated the primary outcome of the study and detailed the measurement methods for all outcomes, along with the statistical tests used (lines 40-43).

Comment 2: "All of benefits that authors mentioned about cesaean section in comparison to normal vaginal delivery, in fact are the hazards of this surgery such as: shorter hospitalization, shorter recovery, better breastfeeding. Please revise.”

Response 2: As requested, we removed the “shorter hospitalization, shorter recovery, and better breastfeeding” (lines 62-63).

Comment 3: "Lines 102-103 needs to be cited.”

Response 3: As per your request, we added relevant citations to support the claims made (lines 104).

Comment 4: "Line 95: please write the reference.”

Response 4: We thank the reviewer for raising this point. This sentence is a concluding statement based on the evidence and reasoning presented earlier in our introduction, where we detail the high cost of processed allografts (lines 86-88), the risk of immunological rejection (lines 88-91), and the logistical simplicity of using autologous tissue available at the point-of-care (lines 87-88). However, to acknowledge the reviewer's valid point that this is a synthesizing claim, we have tempered the language to more clearly present it as our study's conclusion. The revised sentence now reads:

"Based on the factors outlined above, we conclude that in the context of cesarean deliveries, autologous AM may represent a cost-effective, immunologically compatible, and clinically practical alternative to processed grafts." (lines 94-96).

Comment 5: "What was the minimum and maximum score of POSAS? Is there any cut-off?”

Response 5: As per your request, we provided information about the minimum and maximum scores of POSAS and cut-off values. (lines 147-150).

Comment 6: "Statistical analyses: did authors attempt to report effect size?”

Response 6: Since this is a protocol of a study whose results are not ready yet, we cannot report the effect sizes currently. But after the results are ready effect sizes of two-way ANOVA (mean±SD) and logistic regression (odds ratio) will be presented.

Comment 7: "Line 335: why authors wrote that human ethics and consent to participate is NA?”

Response 7: Thank you for raising this issue. We corrected and added an ethical declaration to this section (lines 359-364).

Reviewer #2

We are truly grateful for your kind and encouraging words. Your positive feedback means a great deal to our team and reinforces our commitment to further research in this area. Thank you for taking the time to review our work so thoughtfully.

Reviewer #3

Thank you for taking the time to review our work. We’re grateful for your guidance throughout the review process.

Comment 1: "Thank you for your updated protocol - please note however, that given the breakdown that is given in the response it is possible to unblind the statistician as for ghroups A,B,C,D you are telling us precisely which comparison they are in. So please remove the statement at line 252 as it is not correct.”

Response 1: We removed that statement.

Reviewer #4

We sincerely thank you for your time and valuable feedback.

Comment 1: "The references in the text need to be updated. The manuscript should be re-edited”

Response 1: We have carefully considered all comments and have undertaken a comprehensive revision of the manuscript to address them. We used Grammarly to improve the writing of our manuscript. Also, we added some recently published citations.

Reviewer #5

Thank you for taking the time to share your valuable feedback with us.

Comment 1: "Line 54: The phrase “… CS increased by 12.4% (from 6.7% to 19.1%)” should be edited. First, CS should be spelled out. Second, increased by 12.4% implies a relative increase (RRI), whereas the authors mean an absolute increase (ARI).”

Response 1: We revised the sentence, and it now reads (line 56):

“In the last three decades, CS increased from 6.7% to 19.1% globally.”

Comment 2: "Lines 59–61: The reference used for the sentence is out of date (2012), and I think some of the mentioned advantages (for example, shorter recovery, shorter length of hospitalization, more successful breastfeeding, and control over the birth process) are not correct.”

Response 2: We removed the mentioned advantages (shorter recovery, shorter length of hospitalization, more successful breastfeeding, and control over the birth process) and updated the reference to a meta-analysis published in 2023 (reference No. 6). (lines 62-63).

Comment 3: "The expected recruitment start date registered in the IRCT (2024-07-22) differs from what is reported in the manuscript as the recruitment start date (01/10/2024). Please describe the reason in the protocol.”

Response 3: The reason for the difference is a logistical delay in the initiation of the study. The date registered in the IRCT (2024-07-22) was the projected start date at the time of registration. However, the completion of site preparation tasks was finalized later than anticipated. Therefore, the actual recruitment of the first participant began on 01/10/2024, as reported in the manuscript. To ensure transparency and accuracy, we have added a brief explanation in the 'Methods' section of the manuscript to clarify this. The added text reads:

“While the initial registration in the IRCT indicated a projected start date of 22/07/2024, the actual recruitment of participants began on 01/10/2024 due to administrative finalizations. “ (lines 114-116).

Comment 4: "Line 35: It is mentioned that the trial is double-blind, but the person performing the intervention could not be blinded. According to CONSORT, please specify in the abstract which groups are blinded.”

Response 4: We revised the abstract and added a brief explanation of the blinded parties (lines 35-36).

Comment 5: "It is better not to use uncommon abbreviations such as “PO.””

Response 5: We revised the text and replaced “PO” with “Postoperative”.

Comment 6: "Table 1: It is suggested to use “×” for each assessment time point instead of the current symbol.”

Response 6: We revised Table 1 and replaced “×” with “X”.

Comment 7: "Please report which number was used for the numerator degrees of freedom. I used 3, and the calculated sample size was 141.”

Response 7: We used df = 1 for the calculation. In a 2×2 factorial trial, each effect of interest (main effect of AM, main effect of AF, and their interaction) compares two levels. The numerator df is therefore 1 for each test (AM: 2–1=1; AF: 2–1=1; AM×AF: (2–1)(2–1)=1).

The df = 3 applies only to an overall 4-group ANOVA (testing any difference among the four means), but in a factorial design, the analysis is based on main effects and interaction contrasts, each with df = 1. That is why we used df = 1 in the power calculation and added it in the text (line 293).

Comment 8: "Most of the outcomes will be measured more than once. Please explain how time and the interaction of time × group will be assessed during analysis.”

Response 8: We thank you for this important comment and for emphasizing the need to clarify how time and the time×group interaction will be handled in the analyses.

In this study, outcomes are measured at multiple time points for different practical and clinical reasons, and the importance of “time” varies by outcome. The primary outcome, POSAS, will be analyzed at day 28, which represents the final cosmetic and functional quality of wound healing. The day-10 measurement is collected because patients routinely attend for suture removal at that visit, providing an opportunity to capture early healing data and reduce potential loss to follow-up if some participants do not return at day 28. Since there are only two time points, a longitudinal model would have limited statistical power and add little additional information beyond the final outcome. Therefore, the main analysis will compare POSAS scores at day 28 using two-way ANOVA with factors AM, AF, and AM×AF.

For surgical site infection (SSI), repeated assessments at 36 hours, day 10, and day 28 are intended solely to detect any incident infection within the postoperative period. The timing of onset is not clinically central, and the main parameter of interest is the presence or absence of infection within 28 days. Accordingly, SSI will be analyzed as a binary outcome using logistic regression.

For pain (VAS), which is dynamic and clinically influenced by time, we will conduct a linear mixed-effects model (LMM) including fixed effects for time, AM, AF, and their interactions. If the sample size proves insufficient for stable model estimation, we will instead present two-way ANOVA results at each time point. These clarifications and justifications have been incorporated into the revised Statistical Analysis section.

We added the necessary amendments to the “Statistical Analysis” section (lines 305-326).

Comment 9: "Please describe where all data underlying the findings will be made available when the study is complete.”

Response 9: We revised the “Availability of data and materials” declaration (lines 368-370).

Sincerely,

Dr. Amene Abiri

---

## [Decision Letter · Decision Letter 3]

16 Nov 2025

Effect of Autologous Amniotic Membrane and Fluid on Wound Healing and Complications of Cesarean Section: Study Protocol of A Factorial Randomized Controlled Trial

PONE-D-25-30795R3

Dear Dr. Abiri,

We’re pleased to inform you that your manuscript has been judged scientifically suitable for publication and will be formally accepted for publication once it meets all outstanding technical requirements.

Kind regards,

Ahmed Mohamed Maged, MD

Academic Editor

PLOS ONE

Additional Editor Comments (optional):

Reviewers' comments:

Reviewer's Responses to Questions

**Comments to the Author**

1. Does the manuscript provide a valid rationale for the proposed study, with clearly identified and justified research questions?

Reviewer #1: Yes

Reviewer #4: Yes

Reviewer #5: Yes

2. Is the protocol technically sound and planned in a manner that will lead to a meaningful outcome and allow testing the stated hypotheses?

Reviewer #1: Yes

Reviewer #4: Yes

Reviewer #5: Yes

3. Is the methodology feasible and described in sufficient detail to allow the work to be replicable?

Reviewer #1: Yes

Reviewer #4: Yes

Reviewer #5: Yes

4. Have the authors described where all data underlying the findings will be made available when the study is complete?

Reviewer #1: Yes

Reviewer #4: Yes

Reviewer #5: Yes

5. Is the manuscript presented in an intelligible fashion and written in standard English?

Reviewer #1: Yes

Reviewer #4: Yes

Reviewer #5: Yes

You may also provide optional suggestions and comments to authors that they might find helpful in planning their study.

Reviewer #1: Thanks to authors as they provided comprehensive answers to my previous comments. In my opinion the protocol can be acceptable in the current version.

Reviewer #4: Please use the space provided to explain your answers to the questions above and, if applicable, provide comments about issues authors must address before this protocol can be accepted for publication. You may also include additional comments for the author, including concerns about research or publication ethics.

You may also provide optional suggestions and comments to authors that they might find helpful in planning their study.

I have no opinion on this.

Reviewer #5: The authors addressed all my previous comments. I have no other comment. Congratulation to the authors.

**Do you want your identity to be public for this peer review?** For information about this choice, including consent withdrawal, please see our Privacy Policy

Reviewer #1: No

Reviewer #4: No

Reviewer #5: **Yes: ** Sakineh Mohammad-Alizadeh-Charandabi

---

## [Editor Report · Acceptance letter]

PONE-D-25-30795R3

PLOS ONE

Dear Dr. Abiri,

I'm pleased to inform you that your manuscript has been deemed suitable for publication in PLOS ONE. Congratulations! Your manuscript is now being handed over to our production team.

Kind regards,

on behalf of

Professor Ahmed Mohamed Maged

Academic Editor

PLOS ONE